# MetaPerturb: Transferable Regularizer for Heterogeneous Tasks and Architectures

**Jeongun Ryu**[1*]  **Jaewoong Shin**[1*]  **Hae Beom Lee**[1*]  **Sung Ju Hwang** [1,2]

[1]KAIST, [2]AITRICS, South Korea

{rjw0205, shinjw148, haebeom.lee, sjhwang82}@kaist.ac.kr

## Abstract

Regularization and transfer learning are two popular techniques to enhance model generalization on unseen data, which is a fundamental problem of machine learning. Regularization techniques are versatile, as they are task- and architecture-agnostic, but they do not exploit a large amount of data available. Transfer learning methods learn to transfer knowledge from one domain to another, but may not generalize across tasks and architectures, and may introduce new training cost for adapting to the target task. To bridge the gap between the two, we propose a transferable perturbation, *MetaPerturb*, which is meta-learned to improve generalization performance on unseen data. MetaPerturb is implemented as a set-based lightweight network that is agnostic to the size and the order of the input, which is shared across the layers. Then, we propose a meta-learning framework, to jointly train the perturbation function over heterogeneous tasks in parallel. As MetaPerturb is a set-function trained over diverse distributions across layers and tasks, it can generalize to heterogeneous tasks and architectures. We validate the efficacy and generality of MetaPerturb trained on a specific source domain and architecture, by applying it to the training of diverse neural architectures on heterogeneous target datasets against various regularizers and fine-tuning. The results show that the networks trained with MetaPerturb significantly outperform the baselines on most of the tasks and architectures, with a negligible increase in the parameter size and no hyperparameters to tune.

## 1 Introduction

The success of Deep Neural Networks (DNNs) largely owes to their ability to accurately represent arbitrarily complex functions. However, at the same time, the excessive number of parameters, which enables such expressive power, renders them susceptible to overfitting especially when we do not have a sufficient amount of data to ensure generalization. There are two popular techniques that can help with generalization of deep neural networks: transfer learning and regularization.

Transfer learning [39] methods aim to overcome this data scarcity problem by transferring knowledge obtained from a source dataset to effectively guide the learning on the target task. Whereas the existing transfer learning methods have been proven to be very effective, there also exist some limitations. Firstly, their performance gain highly depends on the similarity between source and target domains, and knowledge transfer across different domains may not be effective or even degenerate the performance on the target task. Secondly, many transfer learning methods require the neural architectures for the source and the target tasks to be the same, as in the case of fine-tuning. Moreover, transfer learning methods usually require additional memory and computational cost for knowledge transfer. Many require to store the entire set of parameters for the source network (e.g. fine-tuning, LwF [21], attention transfer [48]), and some methods require extra training to transfer the source

---

[*]: Equal contribution

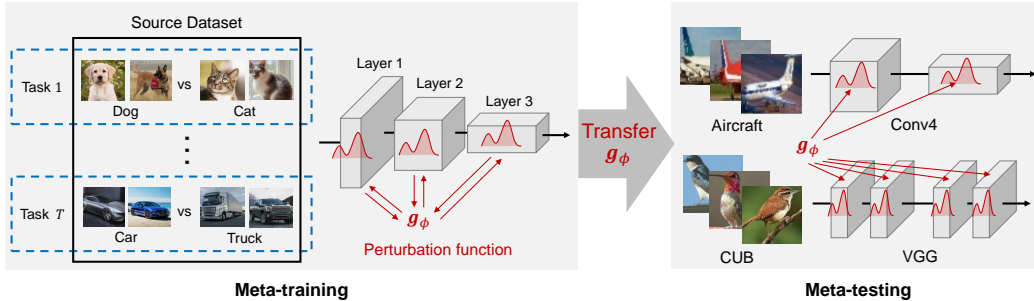

Figure 1: **Concepts.** We learn our perturbation function at meta-training stage and use it to solve diverse meta-testing tasks that come with diverse network architectures.

knowledge to the target task [15]. Such restriction makes transfer learning unappealing, and thus not many of them are used in practice except for simple fine-tuning of the networks pre-trained on large datasets (e.g. convolutional networks pretrained on ImageNet [33], BERT [8] trained on Wikipedia).

On the other hand, regularization techniques, which leverage human prior knowledge on the learning tasks to help with generalization, are more versatile as they are domain- and architecture- agnostic. Penalizing the $\ell_p$-norm of the weights [28], dropping out random units or filters [38, 11], normalizing the distribution of latent features at each input [14, 41, 45], randomly mixing or perturbing samples [50, 42], are instances of such domain-agnostic regularizations. They are more favored in practice over transfer learning since they can work with any architectures and do not incur extra memory or computational overhead, which is often costly with many advanced transfer learning techniques. However, regularization techniques are limited in that they do not exploit the rich information in the large amount of data available.

These limitations of transfer learning and regularization techniques motivate us to come up with a *transferable regularization* technique that can bridge the gap between the two different approaches for enhancing generalization. Such a transferable regularizer should learn useful knowledge from the source task for regularization, while generalizing across different domains and architectures, with minimal extra cost. A recent work [19] propose to meta-learn a noise generator for few-shot learning, to improve the generalization on unseen tasks. Yet, the proposed gradient-based meta-learning scheme cannot scale to standard learning settings which require large amount of steps to converge to good solutions and is inapplicable to architectures that are different from the source network architecture.

To overcome these difficulties, we propose a novel lightweight, scalable perturbation function that is *meta-learned* to improve generalization on unseen tasks and architectures for standard training (See Figure 1 for the concept). Our model generates regularizing perturbations to latent features, given the set of original latent features at each layer. Since it is implemented as an order-equivariant set function, it can be shared across layers and networks learned with different initializations. We meta-learn our perturbation function by a simple joint training over multiple subsets of the source dataset in parallel, which largely reduces the computational cost of meta-learning.

We validate the efficacy and efficiency of our transferable regularizer *MetaPerturb* by training it on a specific source dataset and applying the learned function to the training of heterogeneous architectures on a large number of datasets with varying degrees of task similarity. The results show that networks trained with our meta regularizer outperforms recent regularization techniques and fine-tuning, and obtains significantly improved performances even on largely different tasks on which fine-tuning fails. Also, since the optimal amount of perturbation is automatically learned at each layer, MetaPerturb does not have any hyperparameters unlike most of the existing regularizers. Such effectiveness, efficiency, and versatility of our method makes it an appealing transferable regularization technique that can replace or accompany fine-tuning and conventional regularization techniques.

The contribution of this paper is threefold:

- We propose a lightweight and versatile perturbation function that can transfer the knowledge of a source task to heterogeneous target tasks and architectures.
- We propose a novel meta-learning framework in the form of joint training, which allows to efficiently perform meta-learning on large-scale datasets in the standard learning framework.
- We validate our perturbation function on a large number of datasets and architectures, on which it successfully outperforms existing regularizers and finetuning.

## 2 Related Work

**Transfer Learning**   Transfer learning [39] is one of the popular tools in deep learning to solve the data scarcity problem. The most widely used method in transfer learning is fine-tuning [34] which first trains parameters in the source domain and then use them as the initial weights when learning for the target domain. ImageNet [33] pre-trained network weights are widely used for fine-tuning, achieving impressive performance on various computer vision tasks (e.g. semantic segmentation [22], object detection [12]). However, if the source and target domain are semantically different, fine-tuning may result in negative transfer [46]. Further it is inapplicable when the target network architecture is different from that of the source network. Transfer learning frameworks often require extensive hyperparameter tuning (e.g. which layers to transfer, fine-tuning or not, etc). Recently, Jang et al. [15] proposed a framework to overcome this limitation which can automatically learn what knowledge to transfer from the source network and between which layer to perform knowledge transfer. However, it requires large amount of additional training for knowledge transfer, which limits its practicality. Most of the existing transfer learning methods aim to transfer the features themselves, which may result in negative or zero transfer when the source and the target domains are dissimilar. Contrary to existing frameworks, our framework transfers how to perturb the features in the latent space, which can yield performance gains even on domain-dissimilar tasks.

**Regularization methods**   Training with our input-dependent perturbation function is reminiscent of some of existing input-dependent regularizers. Specifically, information bottleneck methods [40] with variational inference have input-dependent form of perturbation function applied to both training and testing examples as with ours. Variational Information Bottleneck [3] introduces additive noise whereas Information Dropout [2] applies multiplicative noise as with ours. The critical difference from those existing regularizers is that our perturbation function is meta-learned while they do not involve such knowledge transfer. A recently proposed meta-regularizer, Meta Dropout [19] is relevant to ours as it learns to perturb the latent features of training examples for generalization. However, it specifically targets for meta-level generalization in few-shot meta-learning, and does not scale to standard learning frameworks with large number of inner gradient steps as it runs on the MAML framework [9] that requris lookahead gradient steps. Meta Dropout also requires the noise generator to have the same architecture as the source network, which limits its practicality with large networks and makes it impossible to generalize over heterogeneous architectures.

**Meta Learning**   Our regularizer is meta-learned to generalize over heterogeneous tasks and architectures. Meta-learning [14] aims to learn common knowledge that can be shared over distribution of tasks, such that the model can generalize to unseen tasks. While the literature on meta-learning is vast, we name a few works that are most relevant to ours. Finn et al. [9] proposes a model-agnostic meta-learning (MAML) framework to find a shared initialization parameter that can be fine-tuned to obtain good performance on an unseen target task a few gradient steps. The main difficulty is that the number of inner-gradient steps is excessively large for standard learning scenarios, when compared to few-shot learning cases. This led the follow-up works to focus on reducing the computational cost of extending the inner-gradient steps [29, 10, 31, 4], but still they assume we take at most hundreds of gradient steps from a shared initialization. On the other hand, Ren et al. [32] and its variant [35] propose to use an online approximation of the full inner-gradient steps, such that we lookahead only a single gradient step and the meta-parameter is optimized with the main network parameter at the same time in an online manner. While effective for standard learning, they are still computationally inefficient due to the expensive bi-level optimization. As an approach to combine meta-learning with regularization, MetaMixup [24] meta-learns the hyperparameter of Mixup and MetaReg [6] proposes to meta-learn the regularization parameter ($\ell_1$ for domain generalization), but they consider generalization within a single task or across similar domains, while ours target heterogeneous domains. Differently from all existing meta-learning approaches, by resorting to simple joint training on fixed subsets of the dataset, we efficiently extend the meta-learning framework from few-shot learning into a standard learning frameworks for transfer learning.

## 3 Approach

In this section, we introduce our perturbation function that is applicable to any convolutional network architectures and to any image datasets. We then further explain our meta-learning framework for efficiently learning the proposed perturbation function in the standard learning framework.

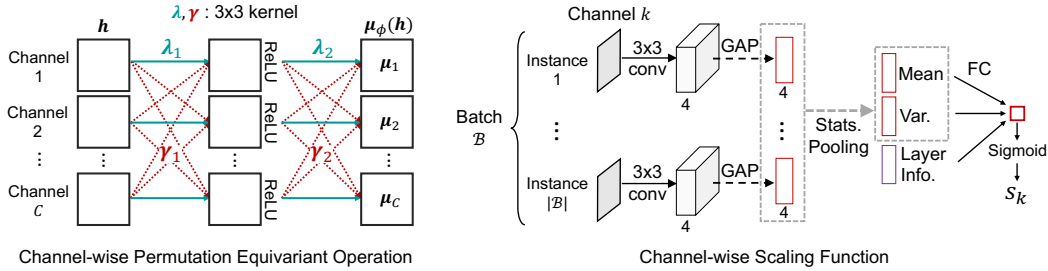

Figure 2: **Left:** The architecture of the channel-wise permutation equivariant operation. **Right:** The architecture of the channel-wise scaling function taking a batch of instances as an input.

## 3.1 Dataset and Network agnostic perturbation function

The conventional transfer learning method transfers the entire set or a subset of the main network parameters $\theta$. However such parameter transfer may become ineffective when we transfer knowledge across a dissimilar pair of source and target tasks. Further, if we need to use a different neural architecture for the target task, it becomes simply inapplicable. Thus, we propose to focus on transferring another set of parameters $\phi$ which is disjoint from $\theta$ and is extremely light-weight. In this work, we let $\phi$ be the parameter for *the perturbation function* which are learned to regularize latent features of convolutional neural networks. The important assumption here is that even if a disjoint pair of source and target task requires different feature extractors for each, there may exist some general rules of perturbation that can effectively regularize both feature extractors at the same time.

Another property that we want to impose upon our perturbation function is its general applicability to any convolutional neural network architectures. The perturbation function should be applicable to:

- Neural networks with **undefined number of convolutional layers**. We can solve this problem by allowing the function to be shared across the convolutional layers.

- Convolutional layers with **undefined number of channels**. We can tackle this problem either by sharing the function across channels or using permutation-equivariant set encodings.

## 3.2 MetaPerturb

We now describe our novel perturbation function, *MetaPerturb* that satisfies the above requirements. It consists of the following two components: input-dependent stochastic noise generator and batch-dependent scaling function.

**Input-dependent stochastic noise generator**     The first component is an input-dependent stochastic noise generator, which has been empirically shown by Lee et al. [19] to often outperform the input-independent counterparts. To make the noise applicable to any convolutional layers, we propose to use permutation equivariant set-encoding [49] across the channels. It allows to consider interactions between the feature maps at each layer while making the generated perturbations to be invariant to the re-orderings caused by random initializations.

Zaheer et al. [49] showed that for a linear transformation $\mu_{\phi'} : \mathbb{R}^C \to \mathbb{R}^C$ parmeterized by a matrix $\phi' \in \mathbb{R}^{C \times C}$, $\mu_{\phi'}$ is permutation equivariant to the $C$ input elements *iff* the diagonal elements of $\phi'$ are equal and also the off-diagonal elements of $\phi'$ are equal as well, i.e. $\phi' = \lambda' \mathbf{I} + \gamma' \mathbf{11}^\top$ with $\lambda', \gamma' \in \mathbb{R}$ and $\mathbf{1} = [1, \ldots, 1]^\top$. The diagonal elements map each of the input elements to themselves, whereas the off-diagonal elements capture the interactions between the input elements.

Here, we propose an equivalent form for convolution operation, such that the output feature maps $\mu_\phi$ are equivariant to the channel-wise permutations of the input feature maps $\mathbf{h}$. We assume that $\phi$ consists of the following two types of parameters: $\boldsymbol{\lambda} \in \mathbb{R}^{3 \times 3}$ for self-to-self convolution operation and $\boldsymbol{\gamma} \in \mathbb{R}^{3 \times 3}$ for all-to-self convolution operation. We then similarly combine $\boldsymbol{\lambda}$ and $\boldsymbol{\gamma}$ to produce a convolutional weight tensor of dimension $\mathbb{R}^{C \times C \times 3 \times 3}$ for $C$ input and output channels (See Figure 2 (left)). Zaheer et al. [49] also showed that a stack of multiple permutation equivariant operations is also permutation equivariant. Thus we stack two layers of $\mu_\phi$ with different parameters and ReLU nonlinearity in-between them in order to increase the flexibility of $\mu_\phi$ (See Figure 2 (left)).

Finally, we sample the input-dependent stochastic noise $\mathbf{z}$ from the following distribution:

$$\mathbf{z} = \text{Softplus}(\mathbf{a}), \quad \mathbf{a} \sim \mathcal{N}(\boldsymbol{\mu}_\phi(\mathbf{h}), \mathbf{I}) \tag{1}$$

where we fix the variance of $\mathbf{a}$ to $\mathbf{I}$ following Lee et al. [19] to eliminate any hyperparameters, which we empirically found to work well in practice.

**Batch-dependent scaling function**    The next component is batch-dependent scaling function, which scales each channel to different values between $[0, 1]$ for a given batch of examples. The assumption here is that the proper amount of the usage for each channel should be adaptively decided for each dataset by using a soft multiplicative gating mechanism. In Figure 2 (right), at training time, we first collect examples in batch $\mathcal{B}$, apply convolutions, followed by global average pooling (GAP) for each channel $k$ to extract 4-dimensional vector representations of the channel. We then compute statistics of them such as mean and diagonal covariance over batch and further concatenate the layer information such as the number of channels $C$ and the width $W$ (or equivalently, the height $H$) to the statistics. We finally generate the scales $s_1, \cdots, s_C$ with a shared affine transformation and a sigmoid function, and collect them into a single vector $\mathbf{s} = [s_1, .., s_C] \in [0, 1]^C$. At testing time, instead of using batch-wise scales, we use global scales accumulated by moving average at the training time similarly to batch normalization [14]. Although this scaling term may look similar to the feature-wise linear modulation (FiLM) [30], it is different as ours is meta-learned and performs batch-wise scaling whereas FiLM performs instance-wise scaling and is not meta-learned.

**Final form**    We lastly combine $\mathbf{z}$ and $\mathbf{s}$ to obtain the following form of the perturbation $\mathbf{g}_\phi(\mathbf{h})$:

$$\mathbf{g}_\phi(\mathbf{h}) = \mathbf{s} \circ \mathbf{z} \tag{2}$$

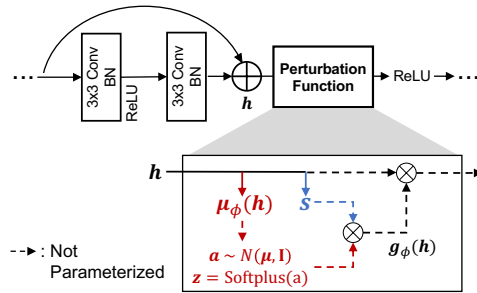

where $\circ$ denotes channel-wise multiplication. We then multiply $\mathbf{g}_\phi(\mathbf{h})$ back to the input feature maps $\mathbf{h}$, at every layer (every block for ResNet [13]) of the network (See Figure 3). We empirically verified that clipping the combined feature map values with a constant $k$ (e.g. $k = 100$) during meta-training helps with its stability since the perturbation may excessively amplify some of the feature map values. Note that since the noise generator is shared across all the channels and layers, our transferable regularizer can perform knowledge transfer with marginal parameter overhead (e.g. 82). Further, there is no hyperparameter to tune[2], since the proper amount of the two perturbations is meta-learned and automatically decided for each layer and channel.

Figure 3: **The architecture of our perutrbation function** applicable to any convolutional neural networks (e.g. ResNet)

## 3.3  Meta-learning framework

The next important question is how to efficiently meta-learn the parameter $\phi$ for the perturbation function. There are two challenges: **1)** Because of the large size of each source task, it is costly to sequentially alternate between the tasks within a single GPU, unlike few-shot learning where each task is sufficiently small. **2)** The computational cost of lookahead operation and second-order derivative in online approximation proposed by Ren et al. [32] is still too expensive.

**Distributed meta-learning**    To solve the first problem, we class-wisely divide the source dataset to generate $T$ (e.g. 10) tasks with *fixed* samples and distribute them across multiple GPUs for parallel learning of the tasks. Then, throughout the entire meta-training phase, we only need to share the low-dimensional (e.g. 82) meta parameter $\phi$ between the GPUs without sequential alternating training over the tasks. Such a way of meta-learning is simple yet novel, and scalable to the number of tasks when a sufficient number of GPUs are available.

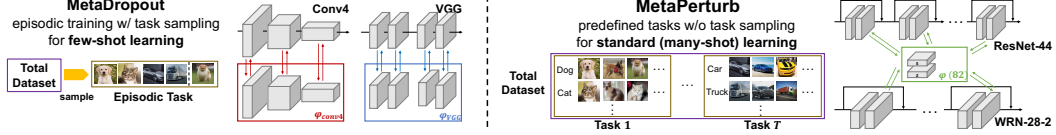

Figure 4: **Comparison with MetaDropout [19].** While MetaDropout also meta-learns a noise generator, it is tied to a specific architecture and cannot scale to large networks and datasets since it learns a different noise generator at each layer and uses MAML, which is not scalable.

| **Algorithm 1** Meta-training | **Algorithm 2** Meta-testing |
|---|---|
| 1: **Input:** $(\mathcal{D}_1^{\text{tr}}, \mathcal{D}_1^{\text{te}}), \ldots, (\mathcal{D}_T^{\text{tr}}, \mathcal{D}_T^{\text{te}})$ | 1: **Input:** $\mathcal{D}^{\text{tr}}, \mathcal{D}^{\text{te}}, \phi^*$ |
| 2: **Input:** Learning rate $\alpha$ | 2: **Input:** Learning rate $\alpha$ |
| 3: **Output:** $\phi^*$ | 3: **Output:** $\theta^*$ |
| 4: Randomly initialize $\theta_1, \ldots, \theta_T, \phi$ | 4: Randomly initialize $\theta$ |
| 5: **while** not converged **do** | 5: **while** not converged **do** |
| 6:     **for** $t = 1$ to $T$ **do** | 6:     Sample $\mathcal{B}^{\text{tr}} \subset \mathcal{D}^{\text{tr}}$. |
| 7:         Sample $\mathcal{B}_t^{\text{tr}} \subset \mathcal{D}_t^{\text{tr}}$ and $\mathcal{B}_t^{\text{te}} \subset \mathcal{D}_t^{\text{te}}$. | 7:     Compute $\mathcal{L}(\mathcal{B}^{\text{tr}}; \theta, \phi^*)$ w/ perturbation. |
| 8:         Compute $\mathcal{L}(\mathcal{B}_t^{\text{tr}}; \theta_t, \phi)$ w/ perturbation. | 8:     $\theta \leftarrow \theta - \alpha \nabla_\theta \mathcal{L}(\mathcal{B}^{\text{tr}}; \theta, \phi^*)$ |
| 9:         $\theta_t \leftarrow \theta_t - \alpha \nabla_{\theta_t} \mathcal{L}(\mathcal{B}_t^{\text{tr}}; \theta_t, \phi)$ | 9: **end while** |
| 10:         Compute $\mathcal{L}(\mathcal{B}_t^{\text{te}}; \theta_t, \phi)$ w/ perturbation. | 10: Evaluate the test examples in $\mathcal{D}^{\text{te}}$ with MC approxi- |
| 11:     **end for** |     mation and the parameter $\theta^*$. |
| 12:     $\phi \leftarrow \phi - \alpha \nabla_\phi \frac{1}{T} \sum_{t=1}^T \mathcal{L}(\mathcal{B}_t^{\text{te}}; \theta_t, \phi)$ | 11: |
| 13: **end while** | 12: |

**Knowledge transfer at the limit of convergence** To solve the second problem, we propose to further approximate the online approximation [32] by simply ignoring the bi-level optimization and the corresponding second-order derivative. It means we simply focus on knowledge transfer across the tasks *only at the limit of the convergence* of the tasks. Toward this goal, we propose to perform a joint optimization of $\theta = \{\theta_1, \ldots, \theta_T\}$ and $\phi$, each of which maximizes the log likelihood of the training dataset $\mathcal{D}^{\text{tr}}$ and test dataset $\mathcal{D}^{\text{te}}$, respectively:

$$\phi^*, \theta^* = \underset{\phi, \theta}{\arg\max} \sum_{t=1}^T \left\{ \log p(\mathbf{y}_t^{\text{te}} | \mathbf{X}_t^{\text{te}}; \text{StopGrad}(\theta_t), \phi) + \log p(\mathbf{y}_t^{\text{tr}} | \mathbf{X}_t^{\text{tr}}; \theta_t, \text{StopGrad}(\phi)) \right\} \quad (3)$$

where StopGrad($x$) denotes that we do not compute the gradient and consider $x$ as constant. See the Algorithm 1 and 2 for meta-training and meta-test, respectively. The intuition is that, even with this naive approximation, the final $\phi^*$ will be transferable if we confine the limit of transfer to *around the convergence*, since we know that $\phi^*$ already has satisfied the desired property at the end of the convergence of multiple meta-training tasks, i.e. over $\theta_1^*, \ldots, \theta_T^*$. It is natural to expect similar consequence at meta-test time if we let the novel task $T + 1$ jointly converge with the meta-learned $\phi^*$ to obtain $\theta_{T+1}^*$. We empirically verified that this simple joint meta-learning suffers from negligible accuracy loss over meta-learning with a single lookahead step [32], while achieving order of magnitude faster training time depending on the tasks and architectures. Further, gradually increasing the strength of our perturbation function $\mathbf{g}_\phi$ performs much better than without such annealing, which means that the knowledge transfer may be less effective at the early stage of the training, but becomes more effective at later steps, i.e. near the convergence. We can largely reduce the computational cost of meta-training with this naive approximation.

**Comparison with MetaDropout** MetaDropout [19] also proposes to meta-learn the noise genera-tor. However, it is largely different from MetaPerturb in multiple aspects. First of all, MetaDropout cannot generalize across heterogeneous neural architectures, since it learns an individual noise gener-ator for each layer (Figure 2 of [19]). Thus it is tied to the specific base network architecture (Fig 4), while MetaPerturb can generalize across architectures. Moreover, MetaDropout does not scale to large networks since the noise generator should be the same size as the main network. MetaPerturb, on the other hand, requires marginal parameter overhead (82) even for deep CNNs since it shares the same lightweight noise generator across all layers and channels. MetaDropout also cannot scale to standard learning with large number of instance and deep networks (Fig 4), since it uses episodic training and MAML for meta-learning. We overcome such a challenge with a scalable distributed joint meta-learning framework described in the earlier paragraphs.

Table 1: **Transfer to multiple datasets.** We use ResNet 20 as the source and target networks. TIN denotes the Tiny ImageNet dataset. The reported results are mean accuracies and standard deviations over 5 meta-test runs.

| Model | # Transfer params | Source dataset | Target Dataset | | | | | |
|---|---|---|---|---|---|---|---|---|
| | | | STL10 | s-CIFAR100 | Dogs | Cars | Aircraft | CUB |
| Base | 0 | None | $66.78_{\pm0.59}$ | $31.79_{\pm0.24}$ | $34.65_{\pm1.05}$ | $44.34_{\pm1.10}$ | $59.23_{\pm0.95}$ | $30.63_{\pm0.66}$ |
| Info. Dropout [2] | 0 | None | $67.46_{\pm0.17}$ | $32.32_{\pm0.33}$ | $34.63_{\pm0.68}$ | $43.13_{\pm2.31}$ | $58.59_{\pm0.90}$ | $30.83_{\pm0.79}$ |
| DropBlock [11] | 0 | None | $68.51_{\pm0.67}$ | $32.74_{\pm0.36}$ | $34.59_{\pm0.87}$ | $45.11_{\pm1.47}$ | $59.76_{\pm1.38}$ | $30.55_{\pm0.26}$ |
| Manifold Mixup [42] | 0 | None | $72.83_{\pm0.69}$ | $39.06_{\pm0.73}$ | $36.29_{\pm0.70}$ | $48.97_{\pm1.69}$ | $64.35_{\pm1.23}$ | $37.80_{\pm0.53}$ |
| **MetaPerturb** | 82 | TIN | $69.98_{\pm0.63}$ | $34.57_{\pm0.38}$ | $38.41_{\pm0.74}$ | $62.46_{\pm0.80}$ | $65.87_{\pm0.77}$ | $42.01_{\pm0.43}$ |
| Finetuning (FT) | .3M | TIN | $77.16_{\pm0.41}$ | $43.69_{\pm0.22}$ | $40.09_{\pm0.31}$ | $58.61_{\pm1.16}$ | $66.03_{\pm0.85}$ | $34.89_{\pm0.30}$ |
| FT + Info. Dropout | .3M + 0 | TIN | $77.41_{\pm0.13}$ | $43.92_{\pm0.44}$ | $40.04_{\pm0.46}$ | $58.07_{\pm0.57}$ | $65.47_{\pm0.27}$ | $35.55_{\pm0.81}$ |
| FT + DropBlock | .3M + 0 | TIN | $78.32_{\pm0.31}$ | $44.84_{\pm0.37}$ | $40.54_{\pm0.56}$ | $61.08_{\pm0.61}$ | $66.30_{\pm0.84}$ | $34.61_{\pm0.54}$ |
| FT + Manif. Mixup | .3M + 0 | TIN | $79.60_{\pm0.27}$ | $47.92_{\pm0.79}$ | $42.54_{\pm0.70}$ | $64.81_{\pm0.97}$ | $71.53_{\pm0.80}$ | $43.07_{\pm0.83}$ |
| **FT + MetaPerturb** | .3M + 82 | TIN | $78.27_{\pm0.36}$ | $47.41_{\pm0.40}$ | $46.06_{\pm0.44}$ | $73.04_{\pm0.45}$ | $72.34_{\pm0.41}$ | $48.60_{\pm1.14}$ |

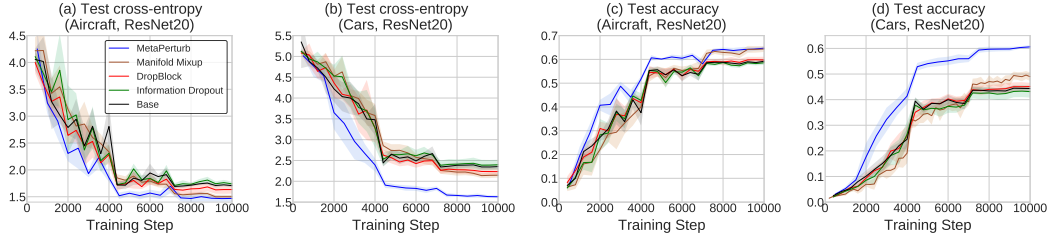

Figure 5: **Convergence plots** on Aircraft [25] and Stanford Cars [17] datasets.

## 4 Experiments

We next validate our method under realistic learning scenarios where target tasks can come with arbitrary image datasets and arbitrary convolutional network architectures. For the base regularizations, we used the weight decay of $0.0005$ and random cropping and horizontal flipping in all experiments.

### 4.1 Transfer to multiple datasets

We first validate if our meta-learned perturbation function can generalize to multiple target datasets.

**Datasets** We use **Tiny ImageNet** [1] as the source dataset, which is a subset of the ImageNet [33] dataset. It consists of $64 \times 64$ size images from 200 classes, with 500 training images for each class. We class-wisely split the dataset into 10 splits to produce heterogeneous task samples. We then transfer our perturbation function to the following target tasks: **STL10** [7], **CIFAR-100** [18], **Stanford Dogs** [16], **Stanford Cars** [17], **Aircraft** [25], and **CUB** [44]. STL10 and CIFAR-100 are benchmark classification datasets of general categories, which is similar to the source dataset. Other datasets are for fine-grained classification, and thus quite dissimilar from the source dataset. We resize the images of the fine-grained classification datasets into $84 \times 84$. Lastly, for CIFAR-100, we sub-sample $5,000$ images from the original training set in order to simulate data-scarse scenario (i.e. prefix *s*-). See the Appendix for more detailed information for the datasets.

**Baselines and our model** We consider the following well-known stochastic regularizers to compare our model with. We carefully tuned the hyperparameters of each baseline with a holdout validation set for each dataset. Note that MetaPerturb does not have any hyperparameters to tune, but there could be variations among runs as with any neural models. Thus we select the best performing noise generator over five meta-training runs using a validation set consisting of samples from CIFAR-100, that is disjoint from s-CIFAR100, and use it throughout all the experiments in the paper. **Information Dropout:** This model [2] is an instance of Information Bottleneck (IB) method [40], where the bottleneck variable is defined as multiplicative perturbation as with ours. **DropBlock:** This model [11] is a type of structured dropout [38] specifically developed for convolutional networks, which randomly drops out units in a contiguous region of a feature map together. **Manifold Mixup:** A recently introduced stochastic regularizer [42] that randomly pairs training examples to linearly interpolate between the latent features of them. We also compare with **Base** and **Finetuning** which have no regularizer added.

**Results** Table 1 shows that our MetaPerturb regularizer significantly outperforms all the baselines on most of the datasets with only 82 dimesions of parameters transferred. MetaPerturb is especially effective on the fine-grained datasets. This is because the generated perturbations help focus on

Table 2: **Transfer to multiple networks.** We use Tiny ImageNet as the source and small-SVHN as the target dataset. As for Finetuning, we use the same source and target network since it cannot be applied across two different networks. The reported numbers are the mean accuracies and standard deviations over 5 meta-test runs.

| Model | Source Network | Target Network | | | | | |
|---|---|---|---|---|---|---|---|
| | | Conv4 | Conv6 | VGG9 | ResNet20 | ResNet44 | WRN-28-2 |
| Base | None | $83.93_{\pm0.20}$ | $86.14_{\pm0.23}$ | $88.44_{\pm0.29}$ | $87.96_{\pm0.30}$ | $88.94_{\pm0.41}$ | $88.95_{\pm0.44}$ |
| Infomation Dropout | None | $84.91_{\pm0.34}$ | $87.23_{\pm0.26}$ | $88.29_{\pm1.18}$ | $88.46_{\pm0.65}$ | $89.33_{\pm0.20}$ | $89.51_{\pm0.29}$ |
| DropBlock | None | $84.29_{\pm0.24}$ | $86.22_{\pm0.26}$ | $88.68_{\pm0.35}$ | $89.43_{\pm0.26}$ | $90.14_{\pm0.18}$ | $90.55_{\pm0.25}$ |
| Finetuning | Same | $84.00_{\pm0.27}$ | $86.56_{\pm0.23}$ | $88.17_{\pm0.18}$ | $88.77_{\pm0.26}$ | $89.62_{\pm0.05}$ | $89.85_{\pm0.31}$ |
| **MetaPerturb** | ResNet20 | $\mathbf{86.61_{\pm0.42}}$ | $\mathbf{88.59_{\pm0.26}}$ | $\mathbf{90.24_{\pm0.27}}$ | $\mathbf{90.70_{\pm0.25}}$ | $\mathbf{90.97_{\pm1.09}}$ | $\mathbf{90.88_{\pm0.07}}$ |

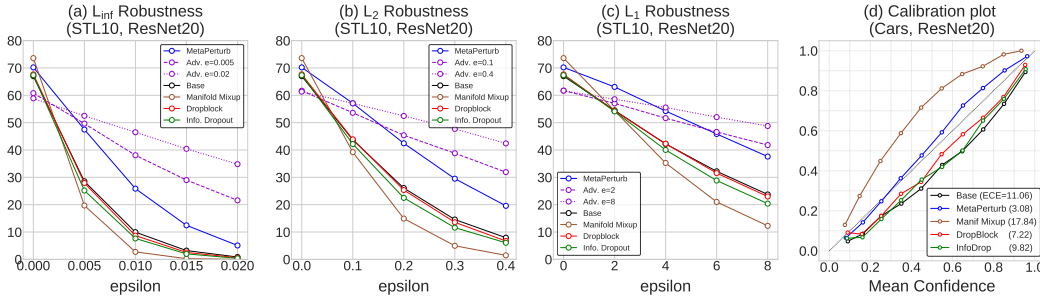

Figure 6: **(a-c)** Adversarial robustness against EoT attacks with varying size of radius $\epsilon$. **(d)** Calibration plot.

relevant part of the input by injecting noise $\mathbf{z}$ or downweighting the scale $\mathbf{s}$ of the distracting parts of the input. Our model also outperforms the baselines with significant margins when used along with finetuning (on Tiny ImageNet dataset). All these results demonstrate that our model can effectively regularize the networks trained on unseen tasks from heterogeneous task distributions. Figure 5 shows that MetaPerturb achieves better convergence over the baselines in terms of test loss and accuracy.

## 4.2  Transfer to multiple networks

We next validate if our meta-learned perturbation can generalize to multiple network architectures.

**Dataset and Networks**    We use small version of SVHN dataset [27] (total $5,000$ instances). We use networks with 4 or 6 convolutional layers with $64$ channels (Conv4 [43] and Conv6), VGG9 (a small version of VGG [37] used in [36]), ResNet20, ResNet44 [13] and Wide ResNet 28-2 [47].

**Results**    Table 2 shows that our MetaPerturb regularizer significantly outperforms the baselines on all the network architectures we considered. Note that although the source network is fixed as ResNet20 during meta-training, the statistics of the layers are very diverse, such that the shared perturbation function is learned to generalize over diverse input statistics. We conjecture that such sharing across layers is the reason MetaPerturb effectively generalize to diverse target networks. While finetuning generally outperforms learning from scratch in most cases, for experiments with SVHN which contains digits and which is largely different from classes in TIN (Table 2), the performance gain becomes smaller. Contrarily, MetaPerturb obtains large performance gains even on heterogeneous datasets, which shows that the knowledge of how to perturb a sample is more generic and is applicable to diverse domains.

## 4.3  Adversarial robustness and calibration performance

**Reliability**    Figure 6(a) shows that MetaPerturb achieves higher robustness over existing baselines against *Expectation over Time* (EoT) [5] PGD attacks without explicit adversarial training, and even higher robust accuracy over adversarial training [23] against $\ell_2$ and $\ell_1$ attacks with small amount of perturbations. Figure 6(d) shows that our MetaPerturb also improves the calibration performance in terms of the expected calibration error (ECE [26]) and calibration plot, while other regularizers do not, and Manifold Mixup even yields worse calibration over the base model.

**Qualitative analysis**    Figure 7 shows the learned scale $\mathbf{s}$ across the layers for each dataset. We see that $\mathbf{s}$ for each channel and layer are generated differently for each dataset. The amount of channel scaling at the lower layers have low variance across the channels and datasets which may be because they are generic features of almost equal importance. Contrarily, the amount of perturbations at the upper layers are highly variable across channels and datasets, since the scaling term $\mathbf{s}$ modulates the amount of noise differently for each channel according to their (noises') relevance to the given task.

Table 3: **Ablation study.**

| | Variants | | s-CIFAR100 | Aircraft | CUB |
|---|---|---|---|---|---|
| | | Base | $31.79_{\pm 0.24}$ | $59.23_{\pm 0.95}$ | $30.63_{\pm 0.66}$ |
| **(a)** | Components of perturbation | w/o channel-wise scaling **s** | $32.65_{\pm 0.40}$ | $63.56_{\pm 1.30}$ | $33.63_{\pm 0.92}$ |
| | | w/o stochastic noise **z** | $31.02_{\pm 0.44}$ | $58.32_{\pm 0.92}$ | $30.26_{\pm 0.67}$ |
| **(b)** | Location of perturbation | Only before pooling | $32.89_{\pm 0.33}$ | $61.39_{\pm 1.01}$ | $38.88_{\pm 1.15}$ |
| | | Only at top layers | $32.57_{\pm 0.46}$ | $57.51_{\pm 0.72}$ | $37.89_{\pm 0.58}$ |
| | | Only at bottom layers | $31.77_{\pm 0.42}$ | $61.32_{\pm 0.29}$ | $33.48_{\pm 0.57}$ |
| **(c)** | Meta-training strategy | Homogeneous task distribution | $34.31_{\pm 0.88}$ | $65.41_{\pm 0.76}$ | $40.64_{\pm 0.31}$ |
| | | MetaPerturb | $\mathbf{34.47_{\pm 0.45}}$ | $\mathbf{65.87_{\pm 0.77}}$ | $\mathbf{42.01_{\pm 0.43}}$ |

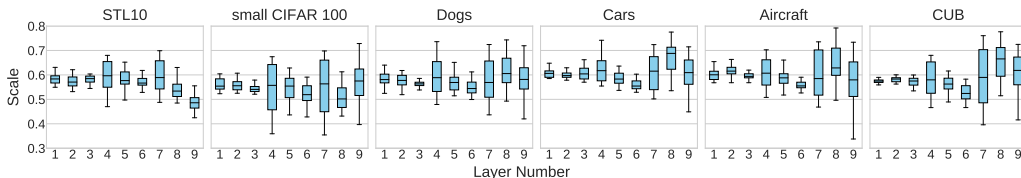

Figure 7: The scale **s** at each block of ResNet20.

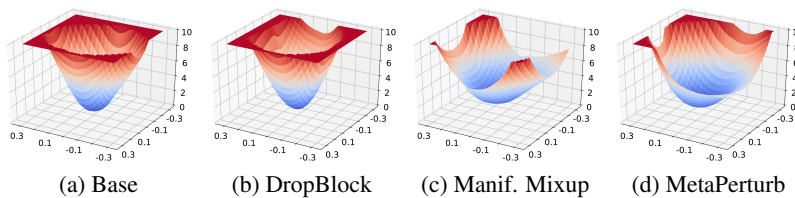

(a) Base     (b) DropBlock     (c) Manif. Mixup     (d) MetaPerturb

Figure 8: **Visualization of training loss surface** [20] (CUB, ResNet20)

Figure 8 shows that models trained with MetaPerturb and Manifold Mixup have flatter loss surfaces than the baselines', which may be a reason why MetaPerturb improves model generalization.

**Ablation study** **(a) Components of the perturbation function:** In Table 3(a), we can see that both components of our perturbation function, the input-dependent stochastic noise **z** and the channel-wise scaling **s** jointly contribute to the good performance of our MetaPerturb regularizer.
**(b) Location of the perturbation function:** Also, in order to find appropriate location of the perturbation function, we tried applying it to various parts of the networks in Table 3(b) (e.g. only before pooling layers or only at top/bottom layers). We can see that applying the function to a smaller subset of layers largely underperforms applying it to all the ResNet blocks as done with MetaPerturb.
**(c) Source task distribution:** Lastly, in order to verify the importance of heterogeneous task distribution, we compare with the homogeneous task distribution by splitting the source dataset across the instances, rather than across the classes as done with MetaPetrub. We observe that this strategy results in performance degradation since the lack of diversity prevents the perturbation function from effectively extrapolating to diverse tasks.

# 5 Conclusion

We proposed a light-weight perturbation function that can transfer the knowledge of a source task to any convolutional architectures and image datasets, by bridging the gap between regularization methods and transfer learning. This is done by implementing the noise generator as a permutation-equivariant set function that is shared across different layers of deep neural networks, and meta-learning it. To scale up meta-learning to standard learning frameworks, we proposed a simple yet effective meta-learning approach, which divides the dataset into multiple subsets and train the noise generator jointly over the subsets, to regularize networks with different initializations. With extensive experimental validation on multiple architectures and tasks, we show that MetaPerturb trained on a single source task and architecture significantly improves the generalization of unseen architectures on unseen tasks, largely outperforming advanced regularization techniques and fine-tuning. MetaPerturb is highly practical as it requires negligible increase in the parameter size, with no adaptation cost and hyperparameter tuning. We believe that with such effectiveness, versatility and practicality, our regularizer has a potential to become a standard tool for regularization.

## Broader Impact

Our MetaPerturb regularizer effectively eliminates the need for retraining of the source task because it can generalize to any convolutional neural architectures and to any image datasets. This versatility is extremely helpful for lowering the energy consumption and training time required in transfer learning, because in real world there exists extremely diverse learning scenarios that we have to deal with. Previous transfer learning or meta-learning methods have not been flexible and versatile enough to solve those diverse large-scale problems simultaneously, but our model can efficiently improve the performance with a single meta-learned regularizer. Also, MetaPerturb efficiently extends the previous meta-learning to standard learning frameworks by avoiding the expensive bi-level optimization, which reduces the computational cost of meta-training, which will result in further reduction in the energy consumption and training time.

## Acknowledgments and Disclosure of Funding

This work was supported by Google AI Focused Research Award, the Engineering Research Center Program through the National Research Foundation of Korea (NRF) funded by the Korean Government MSIT (NRF-2018R1A5A1059921), and the Institute of Information & communications Technology Planning & Evaluation (IITP) grant funded by the Korea government(MSIT) (No.2019-0-00075, Artificial Intelligence Graduate School Program(KAIST))

## Footnotes

[2]The feature map clipping value, $k$, need not be tuned and the clipping could be simply omitted.

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
