[Supplementary Material]

# Supplementary file for MetaPerturb: Transferable Regularizer for Heterogeneous Tasks and Architectures

**Organization**  The supplementary file is organized as follows. In section A, we show additional results and analysis of the robustness and calibration experiments. In section B, we visualize how the perturbations look like in the latent feature space. In section C, we provide the details of the datasets, network architectures, and experimental setups.

Figure 1: **Adversarial robustness** against PGD attack [12] with varying size of radius $\epsilon$ using CUB dataset and ResNet20.

## A  More Results and Analysis on Robustness and Calibration

**Robustness**  In Figure 1 and Figure 6 in the paper, we measure the adversarial robustness of other baseline regularizers such as Manifold Mixup [18], Dropblock [5], and Information Dropout [2]. We use EoT [3] + PGD attack of 200 steps with some range of $\epsilon$ and the inner-learning rate is set to $0.025\epsilon$ for $\ell_\infty$ and $\ell_2$ attack and $0.033\epsilon$ for $\ell_1$ attack. For EoT attack, we sample gradients 10 times. We also compare with *adversarial training* baselines, which take 30 projected gradient descent steps at training. The $\epsilon$ value used for adversarial training for each dataset is written in the Figure 1 and Figure 6 in the paper. We can see that whereas adversarial training is beneficial for the adversarial accuracies, it largely degrades the clean accuracies. On the other hand, our MetaPerturb regularizer improves both clean accuracy and adversarial robustness than the base model, even without explicit adversarial training.

**Calibration**  In the main paper, we showed that the predictions with MetaPerturb regularizer are better calibrated than those of the baselines. In this section, we provide more results and analysis of calibration on various datasets. First of all, calibration performance is frequently quantified with Expected Calibration Error (ECE) [14]. ECE is computed by dividing the confidence values into multiple bins and averaging the gap between the actual accuracy and the confidence value over all the

Table 1: **ECE of multiple datasets.** Source and target network are ResNet20. TIN: Tiny ImageNet.

| Model | # Transfer params | Source dataset | Target Dataset | | | | | |
|---|---|---|---|---|---|---|---|---|
| | | | STL10 | s-CIFAR100 | Dogs | Cars | Aircraft | CUB |
| Base | 0 | None | $23.36_{\pm1.10}$ | $33.09_{\pm0.50}$ | $8.40_{\pm0.66}$ | $9.78_{\pm0.72}$ | $10.37_{\pm0.92}$ | $21.77_{\pm0.80}$ |
| Finetuning | .3M | TIN | $15.68_{\pm0.40}$ | $29.78_{\pm0.33}$ | $11.41_{\pm0.18}$ | $7.00_{\pm0.84}$ | $8.04_{\pm0.65}$ | $23.05_{\pm0.31}$ |
| Info. Dropout [2] | 0 | None | $22.87_{\pm0.28}$ | $32.78_{\pm0.21}$ | $8.27_{\pm0.80}$ | $8.84_{\pm0.77}$ | $9.99_{\pm1.15}$ | $20.41_{\pm0.34}$ |
| DropBlock [5] | 0 | None | $19.65_{\pm0.50}$ | $28.70_{\pm0.17}$ | $5.89_{\pm0.71}$ | $5.83_{\pm1.02}$ | $7.26_{\pm1.55}$ | $18.64_{\pm0.40}$ |
| Manifold Mixup [18] | 0 | None | $5.41_{\pm0.25}$ | $\mathbf{2.26_{\pm0.52}}$ | $5.82_{\pm0.42}$ | $17.00_{\pm0.79}$ | $19.80_{\pm0.45}$ | $\mathbf{9.95_{\pm0.50}}$ |
| **MetaPerturb** | 82 | TIN | $\mathbf{4.80_{\pm0.63}}$ | $14.41_{\pm0.65}$ | $\mathbf{2.05_{\pm0.31}}$ | $\mathbf{2.82_{\pm0.46}}$ | $\mathbf{2.96_{\pm0.37}}$ | $15.62_{\pm1.10}$ |

Figure 2: **Calibration plot** on STL10, s-CIFAR100, Stanford Dogs, Stanford Cars, Aircraft and CUB datasets using ResNet20.

bins. Formally, it is defined as

$$\text{ECE} = \mathbb{E}_{\text{confidence}}\Big[\big|p(\text{correct}|\text{confidence}) - \text{confidence}\big|\Big]. \tag{1}$$

Table 1 and Figure 2 show that MetaPerturb produces better-calibrated confidence scores than the baselines on most of the datasets. We conjecture that it is because the parameter of the perturbation function has been meta-learned to lower the negative log-likelihood (NLL) of the test set, similarly to temperature scaling [6] or other popular calibration methods. In other words, we argue that the learning objective of meta-learning is inherently good for calibration by learning to lower the test NLL.

# B Visualizations of Perturbation Function

In this section, we visualize the feature maps before and after passing the perturbation function from various datasets. We use ResNet20 network for visualization. We visualize the feature maps from the top to bottom layers in order to see the different levels of layers. Although it is not very straightforward to interpret the results, we can roughly observe that the activation strengths are suppressed by the scale **s**, and see how the stochastic noise **z** transforms the original feature maps.

Figure 3: **(a)** Original image **(b-e) Left:** feature map before passing the perturbation **Center:** generated noise **Right:** feature map after passing the perturbation.

## C  Experimental Setup

### C.1  Meta-training Dataset

**Tiny ImageNet**  This dataset [1] is a subset of ImageNet [16] dataset, consisting of $64 \times 64$ size images from 200 classes. There are 500, 50, and 50 images for training, validation, and test dataset, respectively. We use the training dataset for the source training, by resizing images to $32 \times 32$ size and dividing dataset into 10 class-wise splits to produce heterogeneous task samples.

### C.2  Meta-testing Datasets

**STL10**  This dataset [4] consists of 10 classes of general objects such as *airplane, bird*, and *car*, which is similar to CIFAR-10 dataset but has higher resolution of $96 \times 96$. There are 500 and 800 examples per class for training and test set, respectively. We resized the images to $32 \times 32$ size.

**small CIFAR-100**  This dataset [11] consists of 100 classes of general objects such as *beaver*, *aquarium fish*, and *cloud*. The image size is $32 \times 32$ and there are 500 and 100 examples for training and test set, respectively. In order to demonstrate that our model performs well on small dataset, we randomly sample 50 instances per each class from the whole training set and use this smaller set for meta-testing.

**Stanford Dogs**  This dataset [8] is for fine-grained image categorization and contains $20, 580$ images from 120 breeds of dogs from around the world. It has total $12, 000$ and $8, 580$ images for training and testing, respectively. We resized the images to $84 \times 84$ size.

**Stanford Cars**  This dataset [10] is also for fine-grained classification, classifying between the Makes, Models, Years of various cars, e.g. 2012 Tesla Model S or 2012 BMW M3 coupe. It contains $16, 185$ images from 196 classes of cars, where $8, 144$ and $8, 041$ images are assigned for training and test set, respectively. We resized the images to $84 \times 84$ size.

**Aircraft**  This dataset [13] consists of $10, 200$ images from 102 different aircraft model variants (most of them are airplane). There are 100 images for each class and we use $6, 667$ examples for training and $3, 333$ examples for testing. We resized the images to $84 \times 84$ size.

**CUB**   This dataset [20] consists of 200 bird classes such as *Black Tern*, *Blue Jay*, and *Palm Warbler*. It has $5,994$ training images and $5,794$ test images, and we did not use bounding box information for our experiments. We resized the images to $84 \times 84$ size.

**small SVHN (s-SVHN)**   The original dataset [15] consists of $26,032$ color images from 10 digit classes. The image size is $32 \times 32$. In our experiments, we randomly sample $500$ instances per each class from the whole training set for training in order to simulate data scarse scenario. There are $73,257$ examples for testing.

## C.3   Networks

We use 6 networks (Conv4 [19], Conv6, VGG9 [17], ResNet20 [7], ResNet44, and Wide ResNet 28-2 [21]) in our experiments. For Conv4, Conv6, and VGG9, we add our perturbation function in every convolution blocks, before activation. For ResNet architectures, we add our perturbation function in every residual blocks, before last activation.

To simply describe the networks, let `Ck` denote a sequence of a $3 \times 3$ convolutional layer with `k` channels - batch normalization - ReLU activation, `M` denote a max pooling with a stride of $2$, and `FC` denote a fully-connected layer. We provide a implementation of the networks in our code.

**Conv4**   This network is frequently used in few-shot classification literature. This model can be described with `C64-M-C64-M-C64-M-C64-M-FC`.

**Conv6**   This network is similar to the Conv4 network, except that we increase the depth by adding two more convolutional layers. This model can be described with `C64-M-C64-M-C64-C64-M-C64 -C64-M-FC`.

**VGG9**   This network is a small version of VGG [17] with a single fully-connected layer at the last. This model can be described with `C64-M-C128-M-C256-C256-M-C512-C512-M-C512-C512 -M-FC`.

**ResNet20**   This network is used for CIFAR-10 classification task in [7]. The network consists of $3$ residual block layers that consist of multiple residual blocks, where each residual block consists of two $3 \times 3$ convolution layers. Down-sampling is performed by stride pooling in the first convolution layer in a residual block layer and is used at the second and the third residual block layers. Let `ResBlk(n,k)` denote a residual block layer with $n$ residual blocks of channel $k$, and `GAP` denote a global average pooling. Then, the network can be described with `C16-ResBlk(3,16)-ResBlk(3,32)-ResBlk(3,64)-GAP-FC`.

**ResNet44**   This network is similar to the ResNet20 network, but with more residual blocks in each residual block layer. The network can be described with `C16-ResBlk(7,16)-ResBlk(7,32) -ResBlk(7,64)-GAP-FC`.

**Wide ResNet 28-2**   This network is a variant of ResNet, which decrease the depth and increase the width of conventional ResNet architecture. We use Wide ResNet 28-2 which has depth $d = 28$ and widening factor $k = 2$.

## C.4   Experimental Details

**Meta-training**   We use an Adam optimizer [9] and train the model for $2K$ steps. We use a learning rate of $10^{-3}$. We set the mini-batch size to $512$. Lastly, for the base regularizations during training, we use weight decay of $5 \times 10^{-4}$ and simple data augmentations such as random resizing & cropping and random horizontal flipping. In order to efficiently train multiple tasks, we distribute the tasks to multiple processing units and each process has its own main-model parameters $\theta$ and perturbation function parameter $\phi$. After one gradient step of the whole model, we share only the perturbation function parameters across the processes.

**Meta-testing**   We use an Adam optimizer [9] and train the model for $10K$ steps. We use an initial learning rate of $10^{-3}$ and decay the learning rate by $0.3$ at $4K$, $7K$, and $9K$ steps. We set the

mini-batch size to 128. The other configurations are as same as the meta-training stage. After the meta-training is done, only the perturbation function parameter $\phi$ is transferred to the meta-testing stage. Note that $\phi$ is not updated in the meta-testing stage.

**Model selection for transfer learning**    We empirically observed that $\phi$ which maximizes the output feature map works well in the meta-test step. Based on this observation, we select the snapshot of the trained MetaPerturb model at the iteration with the largest average feature map value at the penultimate layer. Moreover, since the performance of our perturbation module may vary across multiple meta-training runs due to stochasticity in the initialization and training, we select the best performing model using a validation set, which is comprised of a subset of the CIFAR-100 dataset, with 100 training instances per class. Note that this validation set does not overlap with the s-CIFAR100 we use in the experimental validation. Although the model selection is not entirely necessary, this may be helpful in practice since we observed that a MetaPerturb regularizer with good performance on a specific dataset consistently works well on any datasets.

**Code**    The code is available at https://github.com/JWoong148/metaperturb.