[Reviews · NeurIPS 2020]

Review 1

Summary and Contributions: The authors propose a new perturbation function which is meta-learnt during the training to improve the generalization on unseen data. Regularization methods don't exploit the large amount of data available while transfer learning may not well generalize to new tasks and architectures. To bridge the gap between two, a meta-learning framework is proposed which jointly learns the perturbation functions over different tasks. Moreover, the proposed approach is evaluated on different architectures and target domains and compare against various regularizations and fine-tuning approaches. Finally, the results verify the effectiveness and generalizability of Metapertub.

Strengths: + bridge gap between Transfer learning and Regularization + easily adaptable to various tasks, architectures, etc, and easy to integrate. + experiments on different target dataset, networks, adversarial robustness etc. + weakness of current regularization and fine-tuning, and shows the limit of these approaches + handle the two different challenges of scalability of meta-learning approach by proposing a more distributed setting of meta-learning as well as knowledge transfer between the tasks by jointly optimize the theta parameters and pertub functions parameters and ignoring the second order derivates.

Weaknesses: - Table1: the results are consistent that Metaperturb improves the fine-grained setting but not the classification, can you please explain this pattern? Same happens in Table 1 of supplementary material. - how do you create Btr and Btest? Is it from the training data or does Btest includes validation set during meta-training? If Btest is from validation set, did you use the examples from validation set in baselines too? - In Meta-testing, did you create the same heterogenous tasks as defined for meta-training in distributed system and then calculate the accuracy per task? How you experimented without having heterogenous tasks at meta-testing stage for a target set? For example, use CIFAR-100 original training set as one single task and compute the accuracy on its original testing set. - Have you tried putting perturb function at the top and bottom layers simultaneously only?

Correctness: Yes, the method looks correct and have empirically shown the effectiveness of it.

Clarity: The paper is easy to read.

Relation to Prior Work: The authors discussed the related work in large and show the shortcomings of the related work by introducing a new Meta-Pertub function which clearly reflects the difference between the proposed one and the previous approaches.

Reproducibility: Yes

Additional Feedback: Q: In figure7, the variation or penalization at the low layers for each target dataset is negligible. Is it because the features are generic for the datasets in those layers?


Review 2

Summary and Contributions: This paper proposed a meta-learning-based framework for perturbation function, which is called MetaPerturb. According to the experiment results, the proposed method can significantly boost more performance over other regularization methods such as dropout and Manifold Mixup.

Strengths: 1. The paper is clear and easy to read, and the proposed method is technically sound. 2. According to the experiment results, the proposed method can improve the classification performance in many settings, especially for fine-grained cases. 3. This paper demonstrates an interesting application of meta-learning algorithm.

Weaknesses: 1. The idea of meta-learning perturbation function is not new. As the authors mentioned, [A] also meta-learns a kind of perturbation. This work seems an incremental version of the previous paper such as [A]. For example, Eq. (1) in this paper is exactly the same as Eq. (5) in [A], except that the forms of the functions µ are different in these two methods. My understanding is that this new paper applies a better noise generator than that of [A]. My view is [A] should also be compared and discussed in Table 1 and Table 2. 2. In Table 1, the proposed method outperforms Manifold Mixup on fine-grained datasets, e.g., Dogs and Cars. However, it fails to achieve better scores on s-CIFAR100 and STL10. The authors also mentioned this point in Section 4.1. As the authors claimed that their method can overcome the problem can transfer knowledge across different domains. I don’t think this claim can hold as the TinyImageNet [B] already contains the same class images (e.g., aircraft). 3. In Table 2, the performance for “Finetuning” is relatively low. There seem to be no improvements compared to the setting “Base”. In previous papers, finetuning can significantly boost performance when transferring the parameters among different domains (e.g., in [C], and [D]). Could you please explain why the experiment results can lead to such a conclusion that is against many previous papers? [A] Lee, Hae Beom, et al. "Meta Dropout: Learning to Perturb Latent Features for Generalization." International Conference on Learning Representations. 2019. [B] Le, Ya, and Xuan Yang. "Tiny imagenet visual recognition challenge." CS 231N 7 (2015). [C] Ye, Han-Jia, et al. "Few-shot learning via embedding adaptation with set-to-set functions." Proceedings of the IEEE/CVF Conference on Computer Vision and Pattern Recognition. 2020. [D] Oreshkin, Boris, Pau Rodríguez López, and Alexandre Lacoste. "Tadam: Task dependent adaptive metric for improved few-shot learning." Advances in Neural Information Processing Systems. 2018.

Correctness: The proposed method seems correct according to the provided experiment results and explanations.

Clarity: Most parts of the paper are written in a clear way. The pseudo codes and figures are easy to understand.

Relation to Prior Work: The authors should compare their method with meeta-learning based perturbation learning methods (e.g., [A]) in the experiment sections. [A] Lee, Hae Beom, et al. "Meta Dropout: Learning to Perturb Latent Features for Generalization." International Conference on Learning Representations. 2019.

Reproducibility: Yes

Additional Feedback: After rebuttal, I feel happy to see authors' responses solving my questions, and so I upgrade my rating to 7.


Review 3

Summary and Contributions: This paper introduces MetaPerturb, a lightweight framework to meta-learn a perturbation function, which improves the generalization capacity of standard architectures and various datasets.

Strengths: I liked the fact that this paper applies meta-learning to an image classification problem which goes beyond the usual few-shot standard. It is also very impressive that they can encode a powerful enough perturbation function using only a handful of meta-parameters (82). The authors also provided the source-code as part of their supplementary materials, which is highly appreciated, along with checkpoints and sample data (which I didn't try myself unfortunately). The code is very clear and easy to read.

Weaknesses: A. Major concerns 1. Lines 277-282: "We see that s for each channel and layer are generated differently for each dataset according to what has been learned in the meta-training stage.". If the weights of the affine transformation that produces s (lines 181-182) is a meta-parameter, then they are shared across all datasets/tasks. It would have been interesting to have a visualization of these weights, in addition to the values of s (since it is hard to understand what are the differences in image statistics from one dataset to another). B. Moderate concerns 1. Lines 181-182: "We finally generate the scales s_1, ..., s_C with a shared affine transformation and a sigmoid function, and collect them into a single vector s = [s_1, ..., s_C] \in [0, 1]^{C}.". Can you comment on the difference between this transformation and (Perez et al., 2017)? 2. Lines 286-287: "It is known that flatter loss surface is closely related to generalization performance, which partly explains why our model generalize well.". I believe this is an unsolved issue (and possibly controversial) in the Deep Learning community. See e.g. (Dinh et al., 2017). C. Minor concerns 1. Line 205 says that \phi is 84 dimensional, whereas line 192 says 82. Is this a typo? 2. Line 205: "we only need to share the low-dimensional (e.g. 84) meta parameter \phi between the GPUs without sequential alternating training over the tasks". Are the gradient wrt. \phi also shared (they have the same dimensionality though)? Are these updates of \phi asynchronous (i.e. some datasets/GPUs are using different versions of the meta-parameter \phi)? (Dinh et al., 2017) Laurent Dinh, Razvan Pascanu, Samy Bengio, Yoshua Bengio. Sharp minima can generalize for deep nets (Perez et al., 2017) Ethan Perez, Florian Strub, Harm de Vries, Vincent Dumoulin, Aaron Courville. FiLM: Visual Reasoning with a General Conditioning Layer

Correctness: The empirical methodology is correct, comparing multiple regularization strategies and different architectures on multiple target datasets. This shows that their method can be applied to a wide variety of settings.

Clarity: The paper is very clearly written.

Relation to Prior Work: As part of the regularization methods, I think (Mai et al., 2019) would be relevant here, where they meta-learn a mixture strategy for Mixup. I also think the paper is missing a crucial reference to FiLM (Perez et al., 2017), which seems very closely related to the framework presented here (not the algorithmic aspect, but the functional aspect). If it is related to FiLM, it is also worth mentioning (Requeima et al., 2019, Bronskill et al., 2020), which also meta-learn the parameters of FiLM (although the latter is a recent paper, which might be considered as concurrent). (Bronskill et al., 2020) John Bronskill, Jonathan Gordon, James Requeima, Sebastian Nowozin, Richard E. Turner. TaskNorm: Rethinking Batch Normalization for Meta-Learning (Mai et al., 2019) Zhijun Mai, Guosheng Hu, Dexiong Chen, Fumin Shen, Heng Tao Shen. MetaMixUp: Learning Adaptive Interpolation Policy of MixUp with Meta-Learning (Perez et al., 2017) Ethan Perez, Florian Strub, Harm de Vries, Vincent Dumoulin, Aaron Courville. FiLM: Visual Reasoning with a General Conditioning Layer (Requeima et al., 2019) James Requeima, Jonathan Gordon, John Bronskill, Sebastian Nowozin, Richard E. Turner. Fast and Flexible Multi-Task Classification Using Conditional Neural Adaptive Processes

Reproducibility: Yes

Additional Feedback: After authors response -------------------------- My concerns were all addressed in the authors response, and I am happy to keep my score unchanged (7).


Review 4

Summary and Contributions: The authors propose a transferable perturbation function (i.e., MetaPerturb) with the aim of transferring the knowledge of a source task to heterogeneous target tasks and architectures. Then, the authors propose a meta-learning framework for efficiently learning the parameters for the proposed perturbation function in the standard learning framework. The authors also evaluate the performance of the proposed perturbation function by training it on a specific source dataset and applying the learned function to the training of heterogeneous architectures 66 on a large number of datasets with varying degree of task similarity.

Strengths: [+] This paper studies a very important problem, i.e., how to improve generalization performance on unseen data. [+] The authors propose a meta-learning framework to perform the joint training over multiple subsets of the source dataset in parallel. [+] The authors conduct experiments to evaluate the performance of proposed method.

Weaknesses: The authors propose a transferable perturbation (i.e., MetaPerturb) with the aim of transferring the knowledge of a source task to heterogeneous target tasks and architectures. Then, the authors propose a meta-learning framework for efficiently learning the parameters for the proposed perturbation function in the standard learning framework. The authors also validate our perturbation function on a large number of datasets and architectures. However, I still have the following concerns. [-] The contribution seems limited. The two proposed components in the perturbation function, i.e., the input-dependent stochastic noise generator and batch dependent scaling function, are derived by directly following the existing works. Specifically, the first component (i.e., the input-dependent stochastic noise generator) is based on two existing works [1,2], and the second component follows [3]. It would be better if the authors give more discussions about their technical contribution over previous works. [-] The authors only consider two techniques for enhancing generalization on unseen data, i.e., transfer learning and regularization. And the motivation is based on the limitations of transfer learning and regularization techniques. However, there also exists other techniques, e.g., data augmentation, adversarial data augmentation. It would be better if the authors give more discussion about these techniques that are used to improve the generalization performance of predictive models. [-] The authors claim that the optimal amount of perturbation can be learned at each layer. However, the authors do not clarify the meaning of the optimality in the context of perturbation settings. Additionally, the authors do not give any theoretical and empirical evidence to demonstrate the optimality of the learned perturbation at each layer. [-] The authors approximate the online approximation by simply ignoring the bi-level optimization and the corresponding second order derivative. It would be if the authors theoretically and empirically analyze such an approximation error. [-] More explanations are needed. Firstly, the authors make an assumption here that the optimal amount of the parameter usage for each channel should be differently controlled for each dataset by using a soft multiplicative gating mechanism. However, the authors do not justify the reasonability of this made assumption. Additionally, the configuration of the hyper-parameters is missing, e.g., the learning rate. [-] The application of the proposed perturbation functions seems limited. The proposed perturbation function is applicable to convolutional network architectures and any image datasets. It would be better if the authors discuss other cases (e.g., neural networks without convolutional layers). [-] The authors fail to cite topic-similar papers, e.g., [4]. It would be better if the authors give further discussion. [1] “Meta Dropout: Learning to Perturb Latent Features for Generalization”, ICLR 2020. [2] “Deep Sets”, NIPS 2017. [3] “Batch Normalization: Accelerating Deep Network Training by Reducing Internal 346 Covariate Shift”, ICML 2015. [4] “MetaReg: Towards Domain Generalization using Meta-Regularization”, NeurIPS 2018.

Correctness: Maybe.

Clarity: Yes.

Relation to Prior Work: More discussions are needed.

Reproducibility: Yes

Additional Feedback:

[Author Response · NeurIPS 2020]



MetaDropout — episodic training w/ task sampling for **few-shot learning** — Total Dataset → sample → Episodic Task — Conv4 — VGG — $\varphi_{conv4}$ — $\varphi_{VGG}$

MetaPerturb — predefined tasks w/o task sampling for **standard (many-shot) learning** — Total Dataset — Dog Cat ... Car Truck ... — Task 1 ... Task $T$ — ResNet-44 — $\varphi$ (82) — WRN-28-2

We thank all the reviewers for constructive comments. Reviewers appreciate that our paper is well-written, clear, and is
tackling the important problem of scaling meta-learning, by proposing a novel distributed framework.
**[Common Comments] Comparison with MetaDropout** [**R2**, **R4**]   Our MetaPerturb is **not** incremental over
MetaDropout [18]. **1) MetaDropout cannot generalize across heterogeneous neural architectures**, since it learns
an individual noise generator for each layer (Figure 2 of [18]). Thus it is tied to the specific base network architecture
(Top Figure), while MetaPerturb can generalize across architectures since it is a size- and order-invariant set function
shared across all layers (L74-75). **2) MetaDropout does not scale to large networks** since the noise generator should
be the same size as the main network. MetaPerturb, on the other hand, requires marginal memory overhead (82
parameters) even for deep CNNs (e.g. ResNet-44, L190-192) since it shares the same lightweight noise generator across
all layers and channels. MetaDropout also becomes almost infeasible to train with large networks due to the needs of
computing the second-order derivatives. **3) MetaDropout cannot scale to standard learning** (Top Figure), since it
uses episodic training and MAML for meta-learning. For standard learning with a large number of instances, taking a
few gradient steps with few sampled instances is highly insufficient for minimizing loss on all instances, and taking
large number of gradient steps over large number of episodes is infeasible. (L115-116) We overcome such a challenge
by proposing a **scalable meta-learning** framework which splits the given dataset into multiple subsets (tasks) without
task sampling, and jointly training the shared set-function across all tasks (L76-77) without lookahead gradient steps.
**Improvement on fine-grained datasets** [**R1**, **R2**]  As mentioned in L252-254, we attribute the improvements to **z** and
**s**, which help focus on the more relevant part of each input, that is crucial for discrimination between two very similar
classes. **Missing references** [**R3**, **R4**]  We will cite them and include the following discussions: FiLM uses instance-
wise modulation whereas our **s** network is a batch-wise set function. MetaMixup meta-learns the hyperparameter
of Mixup and MetaReg proposes to meta-learn the regularization parameter ($\ell_1$ for domain generalization), but they
consider generalization within a single task or across similar domains, while ours target heterogeneous domains.
**[R4] Contributions seems limited.** Please see the comparison against MetaDropout in the general comments. Also,
each component is largely different from the models mentioned: **1) vs. BN:** While BN learns the scaling terms as
free variables, **s** network outputs the scaling factor for each channel as a function of the batch. **2) vs. Deep Sets.** The
DeepSets paper does not deal with channel-wise permutation equivariance for Conv layers, which we newly developed.

**Analysis on approximation error.** We meta-trained MetaPerturb with Ren
et al. [30] with a single lookahead step and meta-test on STL10 for empirical
analysis. The Table on the right shows that Ren et al. [30] increases the
training time by 6× with marginal increase in accuracy. **Why not consider**

| Method | Train time | Accuracy |
|---|---|---|
| MetaPerturb | ~ 1 hr | $69.79_{\pm0.60}$ |
| MetaPerturb w/ Ren et al | ~ 6 hrs | $69.88_{\pm0.50}$ |

**other techniques?** Although there exist diverse approaches to improve generalization, we compared against the most
relevant works (stochastic perturbation) since all other techniques are orthogonal to ours and thus can be used together.
**Jutification of the parameter usage control for each dataset.** Figure 6 shows that the distribution of **s** is different
across the datasets, and the ablation study (Table 3) shows the necessity of the **s** network. **What if it is not CNNs?**
For MLP, perturbation function can be implemented by replacing convolution with linear operations. For RNNs and
Transformers, we leave it as future work. **Missing configurations of hyperparameters.** Please see Section C.4. of the
supplementary file. **Definition of the optimal amount of perturbation.** We will tone down *optimal* to *proper*.
**[R2] TinyImageNet may contain image classes for fine-grained datasets (e.g. aircraft).** TIN contains low-
resolution (32×32) images with general classes (e.g. airplane, bird), while Aircraft and CUB datasets contain
high-resolution images (84×84) and contain fine-grained classes. Thus, we believe that the two datasets are sufficiently
different. **Performance of finetuning.**  In Table 1, finetuning significantly outperforms learning from scratch in all
cases. Yet, for experiments with SVHN which contains digits and which is largely different from classes in TIN (Table
2), the performance gain become smaller. MetaPerturb obtains large performance gains on both cases, which shows that
the knowledge of perturbing a sample is more generic and thus is applicable to diverse domains.

**[R1] Perturb function at the top and bottom layers.**
We performed the suggested experiments, and it per-
forms better than perturbing only the top or the bottom
layer, but is worse than the full model. **Split of $B^{tr}$ and**

| Location of perturb | s-CIFAR100 | Aircraft | CUB |
|---|---|---|---|
| Top layers | $32.54_{\pm0.19}$ | $53.42_{\pm0.79}$ | $27.70_{\pm0.68}$ |
| Bottom layers | $31.75_{\pm0.97}$ | $61.93_{\pm0.86}$ | $31.40_{\pm0.24}$ |
| Top&Bottom layers | $33.63_{\pm0.48}$ | $61.65_{\pm1.65}$ | $32.57_{\pm0.30}$ |
| MetaPerturb | $\mathbf{34.47_{\pm0.45}}$ | $\mathbf{66.12_{\pm0.70}}$ | $\mathbf{39.94_{\pm1.30}}$ |

$B^{te}$? They both come only from the training split of the
original dataset (no fairness issue). **Heterogeneous tasks for meta-test?** At meta-test time, we fix the transferred
perturbation parameters and only train the main model parameters with a single target task.
**[R3] Weight visualization of s network.** We also visualize the weights for the 3x3 Conv
filters and FC layer weight on the right. It shows that the **s** network outputs larger scales
for feature maps with more channels and larger spatial size. **is the gradient of $\phi$ shared?**
Yes, and $\phi$ is updated *synchronously* at every iteration thanks to its small dimensionality
($d = 82$). **Controversial flatter loss surface.** We agree and will tone down the claims.



[Meta-Review · NeurIPS 2020]

This paper introduces a new approach for meta-learning a regularizer that empirically transfers well across different CNN architectures and image datasets. The reviewers agreed that this paper makes a worthy contribution to the NeurIPS community. The authors are encouraged to include the clarifications and new information from the author response in the camera-ready paper, including the clarification on the differences w.r.t. meta-dropout.